# The Sound of One Hand Playing: The Sonic Environment of a Zen Training Temple

**Erez Joskovich**

Department of Philosophy, Ben-Gurion University of the Negev, Beer-Sheva 84105, Israel; joskovic@bgu.ac.il

**Abstract:** Zen practice is often associated with aloof silence and introspection. However, any visitor to a Japanese Zen training temple will be surprised by the abundance and richness of the acoustic environment. In effect, every moment of the training schedule is accompanied, chronicled, and choreographed by percussions and chanting vocals. This paper will introduce this sonic environment and explain how different sounds construct the training experience. In particular, I will focus on sound instruments that coordinate daily activities, such as bells, gongs, drums, and the like, arguing that they are essential to Zen practice. The first part of the paper explains the unique characteristics of a Japanese Zen training temple, focusing on ritual sound instruments and their place within the monastic soundspace. The second part examines the meaning of these instruments as described in Zen canonical writing, focusing mainly on Baizang's monastic regulation and different references to it by Japanese masters throughout history. The third part explores how contemporary monastics understand sound instruments and their function in regulating the body, conveying values, and evoking spiritual transformation.

**Keywords:** Buddhism; Zen; ritual; percussions; training temples; Japan; monastic soundspace

## 1. Introduction

Twenty years ago, when I first entered a monastery, I was struck by the central role of music and ritual performance, which, at that time, was generally overlooked in academic works on the Zen tradition.[1] This was partly due to the challenge of studying the acoustic and auditory aspects of religious experience and the long dominance of philology in the academic study of religion in general, notably Buddhism. However, focusing on the sonic dimension allows us to gain new insights into how religion is expressed, mediated, and communicated. Furthermore, as Pi-yen Chen pointed out, music is a contradictory and adaptable category that can shed light on contemporary studies of Buddhism.[2] Music is not only indispensable in the construction of Buddhist rituals and local cultures, but it is also closely related to the human sensory experience, which has been a primary concern in developing Buddhist traditions (Chen 2001, p. 24).

To my delight, several recent works have examined the place of music within the Japanese Zen tradition. One notable example is Michaela Mross's (2022) work, which combines textual research and fieldwork to examine the importance of recitation within Japanese Zen, most notably the interrelation between the memorization of text, music, and ritual performance. Focusing on the *kōshiki* 講式 ritual, Mross discusses inter alia sutra recitation (*shōmyō* 声明) and the centrality of musical performance within the Sōtō tradition.[3] Another recent example is Joshua Irizarry's (2022), which provides a brilliant ethnographic description of the life and training of contemporary Zen monks in Sōjiji temple (總持寺). While paying close attention to acoustics, Irizarry analysis focuses mainly on the performance of Buddhist hymns (*goeika* 御詠歌).[4] However, regardless of these studies' significant contributions, the sonic experience is not limited to elaborate musical performances of scriptures. In effect, every moment of the monastic training schedule is accompanied, chronicled, and choreographed by various bells, gongs, and percussions.

Accordingly, this paper explores the role of sound-making instruments in the monastic soundspace to understand better their symbolic, mythic, aesthetic, and spiritual significance. Additionally, it is important to note that, while Mross and Irizarry focus on the Sōtō sect, our discussion of the Rinzai tradition will contribute to a more comprehensive understanding of sound within the Zen tradition.

Murray Schafer emphasized the need for a multidisplinarian approach to the study of sound. He stressed the importance of information gathered from various sources to understand past and present soundspace. At the same time, Schafer also pointed to the place of the observer's self-aural awareness in the field (Schafer 2009). Indeed, unlike the study of texts, hearing and producing sounds is a sentinel experience. This rich bodily knowledge cannot be fully transmitted, not even through the many dramatic stories and anecdotes of the Zen tradition. Several scholars have highlighted the significance of sound placement in creating a multisensory experience that is culturally, perceptually, and performatively relevant (see, for examples, Gould et al. (2019) and Hankins and Stevens (2013, especially pp. 5–20)). As such, an ethnographer must undergo an auditory apprenticeship to comprehend the role and significance of sound within a specific context. This process, referred to by Keiko Torigoe as "ear education", particularly applies to monastic soundspace, a unique and distinct environment even for individuals familiar with contemporary Japanese culture (Torigoe (1997) cited in Gould et al. (2019, p. 247)). Accordingly, my two decades of experience in Zen training have been crucial to this study, as it improved my understanding of the consequences of Zen acoustic practices and enabled me to develop what Gould et al. called a "local ear" (Gould et al. 2019, p. 249). In other words, my understanding of the sonic environment in a training temple has developed over time, inspiring my current effort to analyze it through systematic research. Nonetheless, undertaking fieldwork in a Zen training temple in Japan comes with considerable challenges. The monks' hall (*sōdō* 僧堂) is considered a sacred space that is intended to help practitioners reach the ultimate Buddhist goal.

As a result, it is rare for non-monastics, let alone foreign scholars, to be allowed to conduct research there. Even when given access, researchers must comply with the monks' practices and not disrupt their routines. The strict rules and no conversation policy also limit the ethnographers' ability to record their experiences and conduct interviews systematically. Given these challenges, let me specify the particular locations, methods, and considerations for the fieldwork presented in this paper.

I gathered most of the materials for this paper during two terms of fieldwork in Japan. The first was in the summer of 2018 when I conducted a month of fieldwork at Zuiryūji 瑞龍寺, a training temple of the Myōshinji branch (Myōshinji-ha 妙心寺派) of the Rinzai sect located in Gifu Prefecture. During this time, the main ideas presented in this paper started to take shape. In 2022, I spent six months in Japan researching contemporary Zen monastic training. During which, I spent several weeks at Eigenji 永源寺, the principal training temple of the Eigenji branch (Eigenji-ha 永源寺派) located in Shiba Prefecture. These two temples were the only Rinzai training temples, out of a dozen I had contacted, that would allow me to research while staying in the monk's hall. My ability to conduct in-depth interviews within the training temples was not without restrictions, but I did my best to talk to all the monks in training.

Zuiryūji and Eigenji are small temples in rural and secluded areas with only a few monks. Hence, I visited several other training temples to gain a broader perspective. Kamakura's Engakuji 円覚寺 was an important site for this study. Although I could not conduct fieldwork in the monks' hall, Engakuji's size, unique location, and high tourist activity made it a crucial comparison. During my several visits there, I experienced the monastic soundscape, spoke with monks, and interviewed Abbot Yokota Nanrei 横田南嶺 Roshi for valuable insights. To further enhance the validity of my findings, I also visited Myoshinji (妙心寺, Kyoto) and Shōgen-ji (正眼寺 Gifu Prefecture). In June and July 2022, I conducted fieldwork at Chokoku-ji 長谷寺, which included multiple stays of one to three days, during which I participated in monastic practices and interviewed the monks.[5] Despite being a

Sōtō training temple, its unique location and numerous practicing monks provided significant insights into the monastic soundscape. From March to September 2022, I interviewed over twenty priests from the Rinzai and Sōtō sects, asking them about their experiences as monks in training. In some cases, the interviewees shared my insights, while others had different views. Accordingly, this paper aims to investigate the sounds that can be heard in Zen monasteries and present the experiences of those who hear them. By combining insights from fieldwork with classical texts and theories from sensory anthropology and sound studies, this paper constructs the Zen monastic soundscape and examines its geographical, social, cultural, and soteriological functions.

## 2. The Acoustic Landscape

Sound instruments are part of a broad spectrum of natural and artificial sounds comprising the monastic soundspace. To better understand the function of these instruments, we should first introduce the acoustic environment in which they operate. First and foremost, it is essential to realize that the vast majority of Zen temples (*jiin* 寺院) in Japan today are parish temples dedicated mainly to performing funerals and memorial services for their lay parishioners (*danka* 檀家). Even though these temples are the hub of Zen religious activities, which include many ritual sounds and sacred music, these are not considered monasteries in the traditional Buddhist sense. This is because they mainly serve as the residence of a head priest (*jushoku* 住職) and his family, rather than a hermitage for a community of practitioners who live and train according to Buddhist ethical principles.[6] Accordingly, only a handful of Zen temples in Japan today have a sizable community of monks undergoing systematic training under a master.[7] The formal name for such temples is "special training place" (*senmon dōjō* 専門道場 in the Rinzai sect) or special monk's hall (*senmon sōdō* 専門僧堂 in the Sōtō sect).

Most contemporary Zen monasteries, especially those of the Rinzai sect, emerged in their current format during the last 150 years. To exclude minor variations, the practice in these monasteries typically includes meditation, ceremonies, sermons, and manual labor. Regardless of the sect, the daily life of the monks is conducted solemnly and under a strict set of rules and etiquettes, which highly regulates bodily activity.[8] These monasteries are referred to as "thickets" or "groves" (*sōrin* 叢林) and sometimes as "monk's groves" (*sōrin* 僧林) to represent the ideal of a secluded abode where monks are nurtured like trees.[9] It is also customary to call these temples mountains, regardless of their actual topography, once again evoking the idea of a hidden hermitage. While this is still true for many of the monasteries, some major training temples are currently located in highly urban environments, making the sounds of traffic, people, etc. a significant component of their soundspace. The best example is perhaps Sōjiji 總持寺, which is built in the center of Yokohama, the second-largest city and the most populous municipality in Japan.[10] Nevertheless, the temple's buildings and surrounding gardens buffer at least some of the city's hustle and bustle. For example, Chokoku-ji, located off Omote Sandō, is surrounded by a graveyard, making it a (relatively) peaceful island in the heart of Tokyo's bustling metropolis. Accordingly, it is essential to bear in mind that the acoustic landscape can differ depending on the temple's location.

In small training monasteries built in rural areas, nature is essential to the sonic environment.[11] The sounds of wind in the trees, birds tweeting, and water flowing resonate with the sounds of the temple's bells, gongs, and chanting vocals. Each season brings a different soundscape, whether it is the frozen stillness of winter, the rustle of fallen leaves in autumn, the pouring of the rainy season, or the cicadas shrieking in summer. As Schafer suggested, it is a Hi-Fi soundscape with less noise and clearer signals, in which hearing is more peripheral. This hearing is constructed by an open environment where one overhears rather than focuses consciously:

> "*The quiet ambiance of the hi-fi soundscape allows the listener to hear farther into the distance, just as the countryside exercises long-range viewing … From the nearest details*

*to the most distant horizon, the ears operated with seismographic delicacy. When men lived mostly in isolation or in small communities, sounds were uncrowded, surrounded by pools of stillness, and the shepherd, the woodsman, and the farmer knew how to read them as clues to changes in the environment*". (Schafer 1994, pp. 43–44)

Accordingly, natural sounds help create a peaceful, serene atmosphere where ritual sounds resonate and echo doctrinal notions, most notably those of emptiness and original enlightenment, as expressed by the thirteenth-century master Dōgen 道元 (1200–1253):

"*The sounding of the mind must be simply the sounding of emptiness. What we call the sounding of mind is actually the sounding of a bell. If the wind bell does not sound, the mind does not sound. How can we call this the mind's sounds?*". (Leighton and Okumura 2004, p. 69)

According to Michael Fowler, within traditional Japanese spaces such as temples, shrines, or residential Japanese gardens, the distinctions between natural sounds, constructed noise, and music are challenged, as is the difference between sensory modes (Fowler 2015, p. 312). Accordingly, the sounds of everyday life play an essential role in the monastery's aquatics. The sound of footsteps on the wooden floors, the clanging of pots and pans in the kitchen, and the voices of the monks all contribute to the overall soundscape. This is particularly relevant in notable training centers such as Eiheiji and Sōjiji, each hosting dozens, and occasionally even hundreds, of monks. However, one of the most striking aspects of a monastery's sonic environment is actually the absence of certain sounds, notably chatter and electronic devices. Moreover, most Rinzai training temples in Japan today rarely have more than a few monks in practice (*unsui* 雲水). Considering these temples' halls were initially designed to facilitate dozens of monastics, they often sound almost deserted. Such silence amplifies natural sounds and boosts ritual sound instruments.

Moreover, silence plays an integral part in Zen practice, as it stimulates awareness. As eloquently put by John Cage, "[Sounds] that are not notated appear in the written music as silences, opening the doors of the music to the sounds that happen to be in the environment … There is no such thing as an empty space or an empty time. There is always something to see, something to hear. In fact, try as we may to make a silence, we cannot". (Cage [1961] 2011, pp. 67–68). In his understanding of the interconnectedness of sound and silence, Cage seems to echo Dōgen, who argued "*The sound that issues from the striking of emptiness is an endless and wondrous voice that resounds before and after the fall of the hammer*". (Waddell and Abe 2002, p. 14). Accordingly, natural soundspace, artificial sounds, or the absence thereof provide the acoustic background to which the temple's bells and percussion instruments play the rhythm of daily life.

### 3. Sound Instruments and Monastic Regulation

Like most of the procedures in contemporary Zen monasteries, the system of sound instruments can be traced back to Baizhang◎s Pure Rules (Hyakujō shingi, Ch. Baizhang qinggui 百丈清規).[12] The eighth chapter of the codex, Dharma-vessel (Hōgi, Ch. Faqi 法器), is one of the few canonical sources within the tradition to articulate the meaning and usage of these instruments and, as such, is considered the locus classicus for monastic sound instruments, including those still used by Rinzai monasteries today. The chapter commences with the premise that Buddhist teaching is simple and intuitive. Thus, originally, it did not require elaborate ritual music. Moreover, the compiler Dongyang Dehui (東陽德輝, n.d.) argued for the deficiency of ritual music (禮樂), since "*even a song accompanied by tapping on the ground is better than the most skilled musical performance*".[13] This illustration reflects Zen's subitist approach to Buddhist practice, according to which, the mind is identical to Buddha (*sokushin ze butsu* 即心是佛), whether or not one works to polish and purify it through discipline and study.

Dehui argued that, ultimately, there is no need to use ritual sound instruments to realize the teaching. However, similar to other skillful means, sound signals might benefit those of inferior understanding, who are like "*someone who is deaf and dumb*".[14] He counts

three primary functions for the multiple percussions used in a Zen temple: (1) encouraging compliance with the teachings and regulations, (2) guiding those who abide in dark subhuman states, and (3) pleasing gods and humans.[15] To conclude the preface, Dehui articulated the state of mind required for the monk who strikes the sound objects as one of deep meditative absorption (*daijō*, Ch. *dading* 大定), constantly abiding by thusness (*jōjaku*, Ch. *changji* 常寂). Applying such a state of mind while striking will give rise to the Way and ultimately bring about spontaneous transformation. In other words, an attitude of no thought (*mushi*, Ch. *wusi* 無思) or no (intentional) action (*mui*, Ch. *wuwei* 無爲) while striking the sound instrument will enable the original spontaneous conversion promoted by the Zen school. Thus, although Dehui initially stated otherwise, he clearly viewed playing instruments as a way of attending and expressing realization, as long as it is approached with the right attitude.

Next, the chapter introduces the various sound instruments. It describes the primary function of the big bell to awaken the practitioners physically and mentally. Thus, when rang each morning, the bell warns against lingering in sleep, and in the evening, it creates awareness of dark ignorance and reveals one's delusions. Before ringing the bell, the monk should recite the following verse: "*May the sounds of this bell reach beyond the Dharma-realm so that it may be heard by all those in the outer worlds who are obscured by darkness. [May] their hearing perfectly purify them, and they will penetrate realization. [May] all sentient beings achieve perfect enlightenment*".[16] Additionally, it is recommended to visualize Avalokiteśvara, the bodhisattva of compassion, each time one tolls the bell to increase the benefit of its sound. The fourth section of the chapter further develops this association of the bell with Avalokiteśvara.[17] In this section, devoted to the [Buddha] hall bell, the compiler refers to several Indian and Chinese scriptures to elaborate on the bell's power. Thus, it is said that the bell's sound spreads compassion and that its sonic effect can liberate sentient beings and even appease the spirits in hell.

The chapter concludes with two more references highlighting the importance of percussions in Buddhist mythology and doctrine. The first reference is from the third chapter of the Golden-light Sūtra (Ch. *Jingguangming jing* 金光明經), which is also the source of the sutra's name. In this chapter, the Bodhisattva Ruchiraketu dreams of a magnificent golden drum emitting a sublime golden light. This drum stands in the center of a gathering of countless buddhas preaching the Law. When a man in the form of a brahmin strikes the drum, its sounds eradicate the suffering caused by evil states, poverty, difficulties, and fear and carry the power to benefit all sentient beings.[18] The second reference is to the Śūramgama Sūtra (Ch. *Lengyan jing* 楞嚴經), where the Buddha instructs Ānanda: "*Listen carefully to the sounds of the bells and drums from Jetavana that calls the assembly to the midday meal. As the drums and bells resound from one moment to another continuously, what do you think? Do the sounds come to your ears, or do your ears go to the place of the sounds?*"[19] While the first reference emphasizes the miraculous power of sound, the latter koan-like question showcases how sounds can be used to explain the concept of non-duality. This point will be further developed below.

To summarize, the central place of sound instruments within the Pure Rules indicates that they are a significant component of the monastic infrastructure. In addition to resonating with discipline and etiquette, sounds carry symbolic meaning and magical powers. They can minimize delusion and suffering and even awaken the listener. Despite some minor changes and additions, the importance of sound instruments was carried along with the rest of the monastic regulations of the Zen schools from China to Japan.[20] Indeed, from the medieval period onwards, Japanese Zen monks have implemented bells and percussions in their monastic rules.[21] Nevertheless, these regulations rarely elaborate on the meaning of sound beyond what is articulated by the Pure Rules. Arguably, this has to do with the difficulty of describing acoustics using words. More than any other aspect of monastic life, sounds are transmitted as a lived tradition. Accordingly, to further enrich our investigation, let us examine the meaning and function of these instruments within contemporary Zen monasteries.

## 4. Sounding Things

In a Zen monastery, all daily activities are directed using different sound-producing instruments (literally "sounding things", *narashimono* 鳴物). These percussions can roughly be distinguished into six types: (1) bells (*kane* 鐘), (2) drums (*ko* 鼓), (3) boards or planks (*ita* 板), (4) standing bells (*kin* 磬), (5) handbells (*rei* 鈴), and (6) wooden clappers (*shaku*) (see Figure 1). Each of these categories contains instruments of various sizes and made from different materials. The most prominent instrument is undoubtedly the big bell (*ōgane* or *daishō* 大鐘). It typically hangs in a designated tower and is struck by a heavy swinging beam (*shumoku* 撞木). The big bell is rung in the early morning (*gyōshō* 暁鐘) and at dusk (*konshō* 昏鐘) to signal the beginning and end of each training day, as well as on special occasions, such as on New Year's Eve. Its deep, resonant sound is believed to have a calming effect on the mind, and its deep echoes, which roll over the surrounding landscape, have been a preeminent motif in many poems and other literary works throughout history.

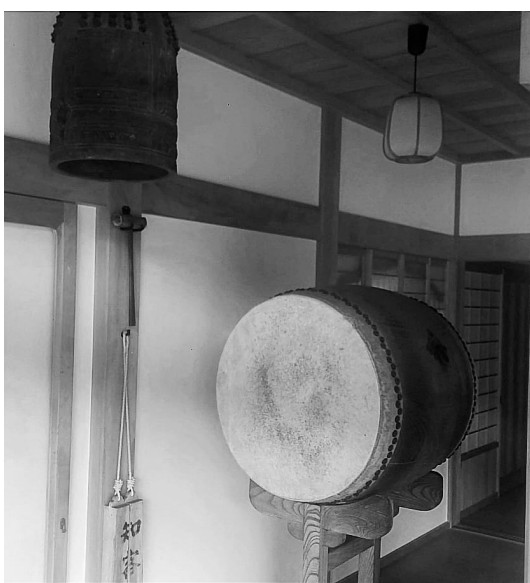

**Figure 1.** Hall bell, drum, and wooden clappers at Eiganji (photographed by the author).

The hall bell (*denshō* 殿鐘) hung in the buddha hall (or main hall, *hondō* 本堂) is used to give monks signals for sutra chanting and other daily services. It is a smaller metal bell struck with a mallet (*kizuchi* 木槌). The wooden plank (*mokuhan* 木板) hung outside the monks' hall is used to signal different times of the day. The cloud plank (*unpan* 雲板) is a flat metal cloud-shaped gong used to announce mealtimes. The handbell (*inkin*) is a small resting bell made of copper mounted on a wooden stick that is struck to direct attention during services and signal the beginning and end of meditation sessions (*zazen* 坐禅). On top of these main categories are unique instruments, such as the fish-shaped wooden drum (*mokugyo* 木魚), which provides the rhythm to sutra recitation, and the summon bell (*kanshō* 喚鐘), which is used to synchronize monks' private interviews with the master (*sanzen* 参禅).

Traditionally, very little explanation is given to trainees in training temples. As I argued elsewhere, the emphasis is mainly on the correct performance of bodily actions rather than the intention or meaning behind them (see Joskovich 2017). It is even more true in the case of sounds, considered part of the infrastructure and, thus, rarely explained. Indeed, I have never heard a master discuss the meaning of sound instruments (or any other ritual, for that matter). Even when I directly inquired about the importance of ritual sounds, my interviewee would often dismiss the question. For example, the late Kyōta Kōnn 清田保南 Roshi, former abbot of Zuiryūji, told me "*These sound instruments (narashimono) have no special meaning … bells, gongs*, etc. *are merely used to give timing and rhythm. They are like a starting gun in a race … they have no special significance whatsoever*". (Private interview (June 2018)). It

is crucial to consider the general attitude of Zen masters, who advise their students against finding hidden meanings or symbolism in everyday life, when interpreting Kyōta Roshi's response. This perspective was shared by many of the interviewees, who refrained from assigning any significance to the ritual sound instruments beyond their practical purposes.

In an interview I had with Rin-san, the head monk of Eigenji, he elaborated on the function of sound instruments as providing signals (*aizu* 合図) for training:[22] "*The sound instruments (narashimono), as their name implies, are meant to signal the monks what is happening. Since personal conversations are not allowed during ceremonies, they give the proper rhythm, indicate what to do in each part, etc. They are used to synchronize between the monks*". (Private interview (August 2022)). Accordingly, the idea that sounds hold the rhythm and harmony of daily life is the most permanent idea regarding the sound instruments' function among contemporary monastics (see, for examples, Hosokawa (2019), Nishimura (1983), and Satō (1972)). This notion is tightly related to Zen practice, as it consists of embodiment through repetition. Rather than understanding or analyzing the Buddhist doctrine, Zen monks constantly repeat movements, gestures, and vocals. They embody Buddhahood through ritual performance, of which sound is a significant component. As argued by Schafer, "*repetition is the memory medium for sound*". (Schafer 2009, p. 34). It is how sounds are retained and explained. The repetitive nature of sound integrates embodiment. Sounds regulate bodily practice, convey values, evoke emotions, and invite the listener to participate not by comprehension but by manifestation.

Inaba Zuiho 稲葉瑞峯, the abbot of Koutokuji (興徳寺, Gifu Prefecture), further expanded upon the function of sound as part of the monastic bodily technique. He attested that, through the use of varied sound instruments, practitioners may cultivate responsiveness. To illustrate this quality, Inaba uses the Japanese phrase (*uteba hiku* 打てば響く), which literally means "ringing when hit".[23] This phrase uses the metaphor of a bell for one's ability to respond quickly. Accordingly, monastic percussions model the trainee's bodily and mental conduct. Constant attention is required for monks to execute timed activities efficiently while contemplating their optimal execution. For instance, when wooden clappers resound before a meal, each clap signals the monks to the next stage in the meal service, i.e., to open their dining set, arrange their bowls in the appropriate sequence, recite a sutra, prepare the food to be served, eat, etc. In this way, using sound instruments transforms daily actions into elaborate choreography.

High proficiency and attention are required not only to respond correctly but also to produce monastic sounds, as described by Irizarry. "*Take, for example, striking the flat bell with the hammer. While sounding simple in theory, the proper form is to hold the hammer with only the thumb and forefinger, creating an axis of motion; the remaining three fingers are used only to hold the long tassel that extends from the end of the hammer to prevent it from flying out of one's hand. The hammer is not used to strike as much as it is made to fall against the flat bell in a controlled manner. Too heavy a fall produces a painful, ear-piercing crack; too light a fall will glance off the bell, producing a muddled sound. Similarly, the handbell is held in an awkward hand position meant to simulate the palms-together gassho; this requires that the entirety of the forearm, and not the fingers, be used to produce a sound. Too much of a flick of the arm will cause the clapper to ring against both the front and the back of the bell, producing two sounds instead of one; too light a flick or holding the bell in a way that tries to use the fingers to control movement, will result in no sound at all*". (Irizarry 2022, p. 175).

Accordingly, playing monastic sound instruments is an essential part of training. It is a skill that the monks are expected to master, just like sutra recitation and other services. It is a physical practice that requires constant attention and repetition in order to be internalized. Moreover, the nature of sound makes it an activity that provides concrete and effective feedback on the monk's proficiency.[24]

The variety of instruments and the complexity of rhythms, particularly those accompanying different services, require much time and effort to master. Even the allegedly simple task of keeping up with all the daily ritual sounds demands considerable work, especially considering most monasteries in contemporary Japan suffer from a workforce

shortage. Moreover, since there are fewer monks, there is less need for coordination, and the time and effort used for complicated rhythms could be better spent on essential tasks like cooking, cleaning, gardening, and attending to visitors, let alone to practice meditation.[25] When I raised the issue of the complexity of playing the instruments with Rin-san, he replied *"Well, I don't know why they are so complicated. But it is decided. It's the rules, and it's universal. People learn it in every age and place, so everyone understands these signals"*. When I suggested simplifying the signals might be more efficient, he rejected the idea: *"Suppose you will change it to something simpler. In such a case, not everyone will be able to understand. It's like language—for example, English or Japanese. If you are in Japan, you need to learn Japanese for people to understand you, and the same with English if you go to America. [These signals] are part of the language of the temple"*. (Private interview (August 2022)).

The comparison between sound signals and language reveals that they have meaning that exceeds their practical use. Indeed, several of my interviewees expressed that sound instruments are not only designed to keep up the daily tempo and rhythm. Yokota Nanrei Roshi, Chief Officer (*kanchō* 管長) of the Engakuji line (Engakuji-ha 円覚寺派)of the Rinzai sect, explained that *"Sounding instruments (narashimono) make people aware of something; they convey messages. Historically, there has always been a connection between music and religion. Within the Rinzai school, the tempo of drumming and other instruments is swift, communicating the message to avoid laziness (daradara) and to approach tasks enthusiastically and quickly. This practice is a part of religious training (shugyō)"*.[26] Several other interviewees similarly referred to the quick tempo of the drums and bells as a means to energize the monks, especially during morning service. The symbolic meaning assigned to the sound instruments is further reinforced by the inscription on the wooden plank (*mokuhan*) that states:

> *"The great matter of life and death*
> *Years and months [passing] are unfortunate*
> *Impermanence is swift*
> *Time waits for no one"*[27]

The hammering of this plank until it wears out further expresses the urgency in diligent practice and the grave matter of wasting time (see Figure 2). Thus, the plank, used to keep and announce the time, is also a ritual symbol of its value. Yet another example is the fish-shaped wooden drum. It is said that, since fish are awake day and night, carving a fish out of wood and striking it is meant to warn people against ignorance and indolence.[28]

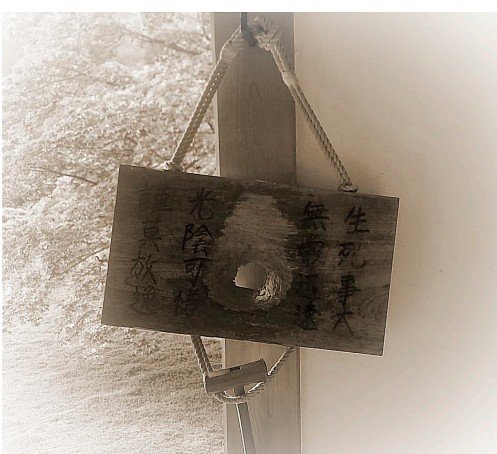

**Figure 2.** A worn-out wooden plank at Eiganji (photographed by the author).

The meaning of sound and, particularly, the tempo they provide to monastic training might be further elaborated by the cognitive approach to the study of music. Many scholars hold that our ability to discern patterns (e.g., melodies, rhythms, and timbres) evokes anticipation that leads to tension and relaxation, which ultimately lock into practices that

give rise to signification, such as mood regulation, bonding, and aesthetic experiences.[29] In line with this approach, Paul Humphreys stressed the importance of the "threefold accelerating roll" common to various sound instruments in multiple contexts throughout the day, notably the wooden plank. This accelerating roll consists of several equi-temporal articulations followed by a full stroke. The first roll is preceded by seven, the second by five, and the third and last one by three. In the Rinzai version, each set of soft, accelerating strokes ends with a robust stroke that rebounds into silence. According to Humphreys, the accelerating roll systematically disrupts the background of temporality (see Humphreys 2004). As one performs or listens to a sequence, their awareness gradually narrows to focus on the specific moments when a particular duration takes place. "*As the field of past-future intentionality becomes narrower with each stroke, the focus of presence becomes deeper. At the moment in which silences of the sequence 'overtake' sounding strokes … the duality of sound and silence collapse into the non-duality of a 'higher order' silence*" (Ibid. 2004, p. 106). Thus, according to Humphreys, not only does the striking of the wooden plank carry doctrinal principles, but it initiates the embodiment of these principles.

One contemporary guide to Zen monastic life wrote "*The sound of the bells is somber and seems to subdue the soul. In ancient times, people heard the Gion temple bells as a reminder of impermanence. Nowadays, what message does the bell ringing throughout the monastery convey to those in the city of Kyoto who hear it from afar?*" Satō (1972, p. 84). The famous opening passage of the Tale of Heike (平家物語, Heike Monogatari) is frequently alluded to in contemporary sources to underscore the profound resonance of the bell, which is said to soothe the spirit and elicit a feeling of transience and remorse.[30] Accordingly, several interviewees emphasized the large bell's emotional, even magical, impact. For example, when I pressed Kyōta Roshi to explain the impact of the monastic bells, he told me a story he had heard on the news several years ago. It was a story about a serial killer who killed eight women and buried their corpses. The police could not locate the weapon or bodies, so they could not establish a case against him. Surprisingly, after hearing the bell from a nearby temple, the killer decided to confess. According to Kyōta, this story suggests that the sound of a large bell can have a profound psychological impact on the listener (Private Interview (June 2018)). It is interesting to note that, on the night of 31 December, Buddhist temples throughout Japan ring their bells in a ritual called Joyanokane 除夜の鐘 or Hyakuhachi shō 百八鐘. During this ritual, the bell is struck 108 times to represent the 108 human afflictions (bonnō 煩悩). Accordingly, the sound of the bell is traditionally associated with popular notions of Buddhist repentance. Kyōta's story suggests that even high-ranked monks—indeed, even Zen masters—hold a multifaceted approach to the impact of sound, especially when it comes to the bell's mystical and purifying properties.

Another notion of monastic sounds as an expression of Buddhist doctrine relates to viewing sounds as an alternative to words in transmitting the essence of the teachings. The central doctrinal trust underlining this argument is the famous Zen slogan of "not relying on words and letters" (*furyōmonji* 不立文字), which asserts that the essence of Zen is directly transmitted from master to disciple rather than via the Buddhist scripture. Thus, the mysterious qualities of sounds are said to communicate and invoke the mind-to-mind transmission (*ishin denshin* 以心傳心) of the Zen doctrine. One could posit that the auditory environment of a Zen training temple plays an integral role in the master's pedagogy, serving as a sonic supplement to their engagement with pupils. In other words, the sonic environment of the Zen temple is not merely of functional value but also a means for spiritual transformation.

This approach finds expression in several Zen koans, directly engaging sound as a medium for awakening, most notably the Sound of the One Hand (*sekishu onjo* 隻手音聲). The famous Japanese master Hakuin Ekaku (白隠慧鶴, 1686–1769), who devised this fundamental koan, explained its meaning as follows:

"*When you clap together both hands, a sharp sound is heard; when you raise one hand, there is neither sound nor smell … It is something that can, by no means, be heard with the ear. If conceptions and discriminations are not mixed within it, and it is quite apart*

*from seeing, hearing, perceiving, and knowing, and if, while walking, standing, sitting,
and reclining, you proceed straightforwardly without interruption in the study of this
koan, then in the place where reason is exhausted, and words are ended you will suddenly
pluck out the karmic root of birth and death and break down the cave of ignorance*".[31]

Similar to the extract from the Śūraṃgama Sūtra cited above, the Sound of the One
Hand plays on the interconnectedness of sound and silence. Rather than preferring silence
over sound, this koan emphasizes the nonjudgmental, non-discriminative listening state.
It points the student to the relationship between silence and sound and between sounds
and the instruments that produce them. The power of sound, especially the connection
between koan and sound, significantly impacted Hakuin, who linked his own awakening
to the sound of a bell:

> "*It was as though I were frozen solid in the midst of an ice sheet extending tens of thou-
> sands of miles. A purity filled my breast, and I could neither go forward nor retreat. To
> all intents and purposes, I was out of my mind, and the Mu* [koan] *alone remained …
> This state lasted for several days. Then I chanced to hear the sound of the temple bell, and
> I was suddenly transformed*". Yampolsky (1971, pp. 117–18)

Several other koans directly refer to the temple's bells—most notably, Stop the Sound
of the [distant] Bell".[32] The accepted reply to this koan is one in which the disciple im-
itates the lingering, resonant sound of the bell (Gongg–).[33] Accordingly, the disciple is
required to manifest non-duality through sound. Their performance is meant to express
the understanding that there is no separation between the instrument producing the sound
and its perceiver. Accordingly, this koan is designed to further cultivate one's non-dual
understanding, as initially expressed in the Sound of the One Hand. Similar to Chen's ob-
servation concerning Buddhist chanting, monks should turn their hearing from pursuing
external sonic objects and listen to their Buddha nature (Chen 2001, p. 37).

Another example from the koan curriculum, which further elaborates on this theme,
is case sixteen of the Gateless Barrier (Zenshū Mumonkan, Ch. Chanzong wumenguan
無門関) Seven-piece Robe at the Sound of the Bell (*shōsei shichijō* 鐘聲七條): "*the earth is so
broad and wide—why do you put on your monk's robe at the sound of the bell?*'" In his comment,
Yunmen further elaborated:

> "*… does the sound come to the ear, or does the ear reach out to the sound? Even if sound
> and silence are both forgotten, how can you understand in words when you reach this
> point? If you use the ears to hear, it is sure to be hard to understand. Only if you sense
> sounds with the eye will you be on intimate terms with Reality*".[34]

As can be seen, Yunmen undermined the logic of words as sounds that convey mean-
ing and suggested that fundamental transformation, or intimacy with reality, comes from
a different kind of listening. In the same tone, Schafer wrote "*Here is a paradox: two things
touch, but only one sound is produced. A ball hits a wall, a drumstick strikes a drum, and a bow
scrapes a string. Two objects: one sound. Another case of one plus one equals one*". (Schafer 2009,
p. 33). Incorporating sound into the koan curriculum provides a unique perspective on
the world. By highlighting the relationship between sound, instruments, and individuals
who listen, this approach challenges traditional views and encourages a deeper under-
standing of the interconnectedness of our surroundings. The examples discussed above
demonstrate the transformative potential of sound in challenging established norms and
fostering an appreciation of no duality.

## 5. Discussion and Further Research

Although rarely emphasized, the soundspace has been essential in constructing the
practice environments of Zen monasteries throughout history. As argued by Douglas
Kahn, a soundspace "*involves actual sonic or auditive events or ideas about sound or listening:
sounds actually heard or heard in myth, idea, or implication; sounds heard by everyone or imagined
by one person alone; or sounds as they fuse with the sensorium as a whole*". (Kahn 1999, p. 3).

Accordingly, the sonic environment of a Zen monastery is a vital aspect of the spiritual practice of its inhabitants. The various natural and artificial sounds regulate activities, resonate tranquility, and echo Buddhist ideas. The complex system of voice signals provides harmony and rhythm, choreographing everyday actions and carrying a ritual and symbolic meaning. Furthermore, these sounds are applied through the koan curriculum to demonstrate non-duality—a critical teaching of the Zen school. As argued by Chen, these sounds possess a function that can direct trainees toward certain specific experiences and allow them to make sense of the coherent relationships between body and mind. Indeed, in a practice space that restricts language, ritual sounds concern much of communal function and institutional meanings (Chen 2001, pp. 24–25).

Hankins and Stevens emphasized the critical role of sound in the constitution of space and community in contemporary Japan. While their collection of papers does not explicitly discuss Buddhism, some of their insights may aid in our understanding of the Zen monastic soundscape. The authors introduced the concept of "sonic practice" to describe the active, embodied practices that imbue sound with meaning. They argued that the significance of sound is not solely derived from its production, transmission, and reception (Hankins and Stevens 2013, pp. 1–20). Rather, they highlighted the embodied ways sounds obtain meaning and how sounds shape individuals or institutions. In this paper, I have shown how sonic practices create monastic life's social and spatial contexts. Sound is integral to monastic life, as it operates within a specific communal, linguistic, and ideological setting. Monks are trained to respond appropriately to various sounds that carry particular meanings and regulate communal practice. As Watarai Sōjun and Sawada Atsuko argued, sound signals transmit monastic rules of conduct through countless reverberations, making them a crucial aspect of monastic discipline and etiquette (see Watarai and Sawada (1980, p. 52), cited in Mross (2022, p. 120)). Novices in Zen monasteries are trained to distinguish between different sound instruments and perform their tasks in accordance with their rhythms. Meanwhile, senior monks can assess the proficiency of their juniors by evaluating their ability to produce sounds accurately. Overall, the training of novices in Zen monasteries is a rigorous process that involves developing a keen sense of hearing and proficiency in producing sounds accurately. As monks progress in their training, they come to appreciate the deeper meanings that sounds can convey, as expressed in koans or Dōgen's teachings.

The *sanzen* service is an example of how sonic practice creates the context in which sound becomes relevant. As mentioned above, several koans, such as the Sound of the One Hand, appeal directly to the auditory dimension of reality. However, using sound instruments is even more critical in contextualizing this practice. A typical koan interview (sanzen) commences with an accelerating roll of strikes on the summon bell (kanshō), indicating that the master room is open for private consultations. Next, the monks line up in a row behind the *kanshō*. After striking the bell once using a malt, each monk rushes toward the master's room to try to confirm his realization of the koan. The master uses a small handbell called *shinrei* 振鈴, similar to the one used to wake up the monks in the morning and, hence, a symbol of awakening. Typically, the master ring indicates that the monk's understanding is insufficient, and he should retreat to allow the next monk in line to enter. The quality of the monk's strike is crucial for the interview's success. If the strike is inaccurate or lacks proficiency, he may be rejected before entering the master's chamber. Accordingly, the *sanzen* is an elaborate ritual performance, a duet between two bells that creates the social and spatial contexts in which koans acquire meaning.[35] As argued by Bernard Faure, a koan, as the oral component of the tradition, might have to do more with the power of a sound than with its meaning (Faure 1991, p. 295).

The study of sound poses a significant challenge due to its intangible nature and the varying contexts in which it is experienced. However, a closer examination of acoustics can yield valuable insights into Buddhist practices, particularly Zen monastic life. While traditional Zen studies focus on textual and visual analyses, delving into theories of materiality and multi-sensoriality, especially concerning the embodied viewpoint of music,

can significantly enhance our understanding of the subject matter. Hankins and Stevens emphasized the crucial role of sonic practices in constructing space. They noted that, in addition to the "absolute" function of space as a container of sound, "*sound both carves out a 'relative' space by setting actors into relation with each other and creates a 'relational' space that characterizes those actors in their relations*". (Hankins and Stevens 2013, p. 5). Thus, creating space through sonic practices requires a nuanced understanding of the unique acoustics of Japanese culture. Japanese scholars have posited that the traditional mode of listening in Japan is particularly distinctive, and this is another area worth exploring (see Gould et al. 2019, pp. 246–48). The Zen monastic soundscape is deeply entrenched in a nostalgic modernity that characterizes present-day Japan.[36] Attention to nature, the passing of time throughout the seasons, and the symbolic significance of the bell are just a few of the cultural factors that contribute to the creation of the monastic space. A thorough analysis of these elements can enhance our comprehension of Zen sonic practices.

As argued by Rosalind Hacket, "*the acoustic design, material, and natural features to optimize the sensual, especially the acoustic, experience of ritual spaces*". (Hackett 2016, p. 316). Consequently, studying monastic architecture and acoustics would provide critical insight into the role of sound and its relation to other sensory experiences, such as sight, touch, and kinesthesia, that comprise the monastic experience. Recent research has established a correlation between sound perception and bodily movements, actions, and environmental interactions.[37] This comprehensive approach incorporates music perception within a broader framework encompassing sensing, movement, cognition, and emotion to elucidate how music impacts human well-being. By integrating physical, cultural, and biological predispositions, we can better comprehend the relationship between sound, doctrine, cognition, and the monastic milieu. Although there may be difficulties in acquiring empirical data, I believe that the embodied musical cognition framework can provide novel insights into the role of sound in Buddhist training.[38]

**Funding:** This research received no external funding.

**Institutional Review Board Statement:** Not applicable.

**Informed Consent Statement:** Informed consent was obtained from all subjects involved in the study.

**Data Availability Statement:** Data sharing is not applicable to this article.

**Acknowledgments:** I would like to express my sincere gratitude to the Japan Foundation for awarding me a research grant that allowed me to conduct my fieldwork in Japan in 2022. I am also deeply thankful to Yamabe Nobuyoshi at Waseda University for his warm hospitality. I extend my appreciation to Rev. Mori, Kyōta Roshi, Yokota Roshi, Rev. Jiun, Rev. Ogawa, and all the other monks and nuns who generously hosted me at their temples and shared their knowledge and insights with me. Additionally, I am grateful to Barbara Wall for inviting me to present an early version of my paper at the AAS Conference in Boston in March 2023 as part of the panel "Buddhist Sounds in Contemporary Japan and South Korea" and to the rest of the panel members for their valuable feedback and advice.

**Conflicts of Interest:** The author declares no conflict of interest.

## Abbreviation

T = Taishō Shinshū Daizōkyō 大正新脩大藏經

## Notes

[1] My interest matured into two papers discussing the role of ritual within the Rinzai tradition (see Joskovich 2017, 2019).

[2] It is beyond the scope of this paper to discuss the doctrinal and historical status of music in Buddhism, as several scholars have broadly considered this topic (see, for example, Mabbett (1993) and Chen (2001)).

[3] In Japan, chanting Buddhist prayers is known as *shōmyō*, which encompasses a wide range of styles. Each school of Buddhism has its unique *shōmyō* tradition, and certain schools have subschools of *shōmyō* that align with sectarian divisions. *Shōmyō* is differentiated from the mere recitation of Buddhist sutras by its elaborate melodies and unique embellishments (see Nelson 2003). (As this paper focuses on Japanese Zen, all transliterations are in Japanese unless otherwise mentioned).

4    In fairness, Mross discusses musical instruments in the monastic regulation (Mross 2022, pp. 120–28). Likewise, Irizarry's description of monastic life often mentions the use of sound instruments (see, for example, Irizarry 2022, p. 175). Accordingly, this paper aims to expand their discussion to the Rinzai tradition and explore different theoretical approaches to the monastic soundspace.

5    Chokoku-ji serves as an urban extension of Eiheiji (永平寺, Fukui Prefecture), one of the two major training temples of the Sōtō sect.

6    Buddhist monks in Japan have been officially permitted to marry and have children since the nineteenth century. As a result, most of the temples are passed down from the priest to his oldest son, making them hereditary.

7    There are approximately sixty operational training temples in all of Japan, out of more than 20,000 Zen temples (see Foulk 2005, pp. 156–70). For an updated list of training temples of the Rinzai sect, see http://shinden.boo.jp/wiki/ 臨済宗専門道場, (accessed on 20 September 2023) and for the Sōtō sect, see https://www.sotozen-net.or.jp/organization/sodo-list (accessed on 20 September 2023).

8    In this paper, I use the term "monks" to refer to both female and male monastics. The reason for this is twofold. Firstly, the monastic training of both monks and nuns is quite similar. Secondly, there are very few nuns in the Rinzai tradition, with only one female training monastery and not even a single female master. Even though the number of nuns in the Sōtō sect is relatively higher, they still constitute a minority.

9    This name derives from the Sanskrit term *pindavana*, used in India for an assembly of mendicants who practiced in the forests (see Zengaku Daijiten 1978, 757d).

10    For a detailed description of Sōjiji's soundspace, see Irizarry (2022, pp. 1–7).

11    It is important to note that the distinction between Rinzai and Sōtō training temples is not always clear-cut. Some training temples belonging to the Sōtō sect, such as Sōbōji (正法寺, Iwata Prefecture), are small and located in rural areas. On the other hand, there are some prominent Rinzai training temples, like Myōshinji (妙心寺, Kyoto), that are situated in urban areas. Accordingly, the following observations are correlated to the type of temple rather than its sectarian affiliation.

12    Zen monastic regulations can be traced back to the influential Tang dynasty Chan Master Baizhang Huaihai 百丈懷海 (720–814). However, the text known as Chixiu Baizhang qinggui 敕修百丈清規 was compiled by Dongyang Dehui in 1335 CE at the request of the Yuan Court, allegedly based on the original regulation by Baizhang, which did not survive (see Foulk 2004, pp. 280–83).

13    擊壤之歌不如九成之奏 (T 48, 2025: 1155b10).

14    若聾瞽焉 (T 48, 2025: 1155b10-14).

15    叢林至今倣其制而用之。于以警昏怠。肅教令導幽滯而和神人也 (T 48, 2025: 1155b17).

16    願此鍾聲超法界。鐵圍幽暗悉皆聞。聞塵清淨證圓通。一切衆生成正覺 (T 48, 2025: 1155b26-27).

17    It is important to note that Avalokiteśvara is commonly translated as "the one who observes the sounds of the world" (Kanzeon, Ch. Guanshiyin 觀世音).

18    See (T 16, 663: 336b11-21). For an English translation, see Emmerick (1970, pp. 8–9).

19    See (T 48, 2025: 1156b10-12). A similar passage is also found in the Lotus Sūtra (T 19, 945: 115c13-24).

20    For example, the small hand bell (*inkin*引磬) did not exist in Song China, and some instruments were given different names, for example, the wooden clappers (*taku* Ch. *tuo*柝) are often referred to as *shaku* (尺) in Rinzai training temples (see Honda 2019, pp. 37–38).

21    See, for example, Dōgen's Eiheishingi, the foundation of contemporary Sōto sect temple regulations (永平清規, T 2584), and The Abbreviated Shō sōrin ryakushingi 小叢林略清規, compiled by Mujaku Dōchū (無著道忠, 1653–1744) in 1684, which form the foundation of contemporary Rinzai monastic rules. Both texts devote large sections to the different sound objects and the appropriate ways to play them.

22    In Zen monasteries, monks in training are often referred to by an abbreviation of their Buddhist name with the suffix san. Thus, for example, Bankei 盤珪 would be Ban-san or Mr. Ban.

23    See https://www.myoshinji.or.jp/houwa/archive/children/020 (accessed on 20 September 2023).

24    Learning to play traditional sound instruments in the Zen tradition can be intricate, as demonstrated by modern instructional manuals created to educate monks and nuns. The Rinzai and Sōtō sects offer these manuals, often with an accompanying CD or audio file showcasing different sutra recitation bells and percussion rhythms. Typically, these instruments are utilized by priests during ceremonies held for their lay followers (see, for example: Gōkohōshiki bonbai shō江湖法式梵唄抄. Yoshitada Yoshitada 吹田良忠 ed. Zen Bunka kenkyūjo禅文化研究所Rinzai kyōten kenkyūkai臨済宗経典研究会. 2018 and Sōtō-Zen: Chōka, senbō, narashimono 曹洞禅: 朝課 懺 法 鳴物. Tōshiba EMI THX–90055, 1980, 6 LP records. Booklet by Watarai Shōjun 渡井正純 and Sawada Atsuko 澤田篤子).

25    This statement is meant as sarcasm. In a Japanese training temple, every action is considered practice, and the priority of *zazen* is far less central than it is often presented in the West.

26    Private interview (September 2022).

27    生死大事 光陰可惜 無常迅速 時不待人.

28     T 48, 2025: 1156a04.

29     A prominent paradigm in the research on music cognition focuses on the anticipation of perceived structural components (see, for examples, Honing 2011 and Huron 2006).

30     The opening passage of the Tale of the Heike states "*The sound of the Gion Shōja bells echoes the impermanence of all things; the color of the sāla flowers reveals the truth that the prosperous must decline. The proud do not endure, they are like a dream on a spring night; the mighty fall at last, they are as dust before the wind*". (translated by McCullough 1988).

31     Translated by Yampolsky (1971, p. 164) (with minor changes). The Sound of the One Hand is considered a fundamental koan for realizing the Dharma-body (*hōshin* 法身) according to Hakuin◎s koan system. Following Hakuin, it became a First Barrier (*shokan* 初關) koan in the Japanese Rinzai school.

32     This koan, included in the Miscellaneous Part (*zatta no Bu* 雜則の部) of the koan curriculum, has several versions. The Myōshinji line version is "*Show me how [you] stop the sound of the bell* (鐘の鳴る音を止めてみよ)". Other lineages use slightly different versions, such as "*Stop the sound of the distant temple bell*". (see Hirotaka 2014, p. 76).

33     This reply is presented in the famous *Gendai sōjizen hyōron* (lit. "a critique of present-day pseudo-Zen") compiled in 1916 by Tominaga Shūho (冨永秀甫) under the pseudonym Hau Hōō (破有法王; lit. "The dharma king destroyer of the bonds of existence") (see Hau ([1916] 1971, p. 99) and Hoffmann (1975, p. 73) for English translations).

34     T 49, 2005: 295a17-19.

35     On koan as a ritual performance (Stephenson 2005).

36     On nostalgia in modern Japan, see Ivy (1995) and within Japanese religion, see Reader (1987).

37     A concise introduction to this approach can be found in Leman and Pieter-Jan (2014).

38     Based on my experience, training temples are not very keen on research, and even if they allow scholars, they ensure that their presence does not interfere with the practices of the monks. Therefore, it is unlikely that they would permit the placement of technical equipment such as microphones, stands, cables, and so on. I had to rely on my smartphone for most of my recordings, which did not yield optimal results.

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
