# Peer review of "The Sound of One Hand Playing: The Sonic Environment of a Zen Training Temple"

_religions, doi:10.3390/rel14101246_

Round 1
Reviewer 1 Report
This is basically an interesting and well-written paper on a relatively new and thought-provoking topic that will help readers gain a better understanding of the sonic experience in Zen ceremonial culture and everyday perceptions, with an emphasis on the role of musical instruments and sound devices at contemporary temples in light of classical monastic regulations. However, there are some fundamental weaknesses in the scope and documentation of the research, and its lack of engagement with many of the appropriate or necessary sources and scholarly materials; this is outlined below.
Therefore, while I recommend that the author do additional research and revise the paper, I also suggest that the revisions should not be a simple process with a quick turnaround. Instead, the author should take their time to explore a variety of new resources and then recast and rewrite the arguments with these materials firmly in mind and well used.
The following is a brief summary of what is lacking in the current version of the paper:
1. In the first paragraph on lines 25-27, the author misrepresents two crucial terms, shomyo and bonbai, while ignoring other key terms. The author mentions two extremely important books by Mross and Irizarry, but in an almost dismissive way and without giving any indication the author has read carefully or engaged seriously with these works. This lack highlights right from the start a major weakness in the article.
2. There are other examples of key works that are cited in the bibliography but are not used or are barely mentioned in the body of the paper.
3. Additionally, there are important works in English by Steven Nelson, Ouchi Fumi, Ho Li-Hua, and Stephen Slottow that should be read and engaged with by the author.
4. The author does not cite or use enough recent Japanese scholarly sources, and may not be aware of important studies in the field by Watarai Shojun, Sawada Atsuko, and Ozaki Shozen.
5. Even more importantly, there is a variety of manuals on music and sound instruments used in contemporary Zen temples in Japan that the author does not cite and may not be familiar with. Ideally these could have been collected during the author's fieldwork.
6. The author relies heavily on fieldwork, which is very good in principle, but apparently without giving much thought to the strictures of methodological applications and controls. There should be a much better documentation of how and why these temples were selected and what time period the author spent studying and practicing there. Also, how and why were the informants chosen, and why should readers assume that a few "private interviews" are authoritative, especially when it is acknowledged that specificity was often lacking. Without more attention paid to clarifying these points, the paper begins to seem more like a journalistic description of some experiences and reflections with a focus broader scholarship on sacred sounds, rather than a tightly reasoned and solid academic analysis of Japanese Zen.
7. The author cites extensively some important classical Zen monastic regulations and sayings that are significant and relevant to the topic, but the author also skips over the late medieval and early modern periods, especially the function and development of "shingi" texts and rites in Japanese Buddhist history. If the argument to be made in the paper is based on linking premodern texts with modern perceptions, it is necessary to provide some consistency and continuity in the overall approach.
8. A minor point is that the author is inconsistent in sometimes using the macron for long vowels while usually using "ou," which is not fitting for a scholarly article. On line 370 "monjin" should be "monji".
In sum, I would like to encourage the author to pursue and polish their research on an exciting theme, but also to point out that this would need to take some time for the scholarship to engage deeply with a fuller range of materials.
Author Response
Thank you for giving my paper a thorough review and for the insightful comments. I have made all the necessary changes to the manuscript based on their suggestions. Below are the significant adjustments I made:
- To clarify how the paper's argument refers to Mross (2022) and Irizarry (2022), I expanded my discussion of these sources in the Introduction (lines 31-48 and fn. 5). I also engaged these two sources in Section 4 (lines 320-332) and several footnotes throughout. Additionally, fn. 4 clarifies the meaning of shōmyō
- I further developed my engagements with Schaefer 1994 (lines 255-302), Gould et al. (lines 60-63, and 536-540), and Faure 1991 (lines 809-811).
- Due to the scope of this research, I cannot engage with all critical works written on the topic
- I added references to several Japanese scholarly works, including by Torigoe Keiko (lines 57-59), Watarai Sōjun, and Sawada Atsuko (lines 500-502) .
- Additionally, contemporary music and sound instrument manuals were also included
- I extensively revised the Introduction to include a broad discussion of the ethnographical work, which includes the locations presented and the consideration entailed in choosing them (lines 136-201).
- While I agree with the importance of discussing the development of musical instruments in late medieval and early modern Zen monastic regulations, it exceeds the scope of this paper
- I have also reviewed the entire text and made necessary corrections based on the reviewers’ suggestions. This includes copy editing the Japanese transliteration
Reviewer 2 Report
Overall this paper is well written, and it was enjoyable and informative to read. It also presents an interesting focus on the sonic sound and their significance in Zen Buddhist practice, which is significant. With some minor improvements and extension of the theoretical analysis this paper will make an academic contribution to the field.
Some suggestions to further enhance the quality of the argument made:
The overall argument could be enhanced by extending the theoretical analyses in several places.
Firstly, more in-depth engagement with the notion of the sacred is necessary. Currently, the author takes this concept/notion for granted. What is meant by "sacred music", why not just use the term music? Do the monks/trainees use such terminology to describe the sounds? Why describe it as sacred? What is mean by the sacred in this context? If the monks use this terminology what do they mean by it? Etc. Engaging more in-depth conceptually with the term sacred would enhance the central argument about how sonic sounds affect a different embodied state (outside the ordinary?).
Related to this: in Line 40 & 41 The author states the paper focuses on the ritual sound instruments used to accompany and coordinate daily activities in a training temple and that the study of religion has prioritised sight over other senses-- here the author may want to take into account the various new development in affect theory, or the focus on sonic, embodied practices in fro example the volume Sound, Space, and Sociality in Modern Japan edited by Joseph Hankins and Caroline Stevens (2014). While the examples and contexts in this volume differ, the author could think about what is similar about the affect of sonic embodied experience, linking this up to why this is significant in the Zen monastery as discussed in detail by the author. Linking the article to such wider debates within anthropology will enhance the argument.
There are obviously various notions of what an enlightened consciousness mean, and more precisely acknowledging the specificity of the notion of enlightenment in the Zen context would be more accurate. Furthermore, what does enlightenment means here apart from a more general notion of non-duality could be further explored and engaged with more critically in light of ethnographic details from those interviewed/talked to. How does it differ from other sonic embodied experiences? Although this is not the main argument, it does link to what is supposedly affected by sonic experience.
The overall discussion would be enhanced by adding somewhat more analyses in various places. For example, Line 363-4 what does this anecdote tell us about the person telling it? Why is that seen as significant to this interlocutor? What does this tell us? Etc. More examples, could be given from fieldwork. Or, in Line 425-6 extend the analysis; Or, in Line 453-4 expand on what is meant by "sound creates sacred space" - elaborate (in connection with previously suggested point about the 'sacred').
The English quality is fine, apart from one sentence in Line 447-9, which is somewhat convoluted and could be improved. This is an important concluding point, so expanding clarity here is important.
Author Response
Thank you for giving my paper a thorough review and for the insightful comments. I have made all the necessary changes to the manuscript based on your suggestions
I enhanced the discussion part in Section 5, where I engaged with Hankins and Stevens 2013 (lines 763-829) and added more analyses throughout the paper
Reviewer 3 Report
Excellent contribution.
Twice appears the word "belles," unless this is special use of the plural for bells, if should be fixed.
The sentence on lines 226-228 is awkward, the use of "motive."
The sentence on line 177, does the "may" fit with "will"?
Line 309: "these" should be "These
Line 356: needs end quote (")
377: "awaking" should be "awakening"
Line 412: "we" then "your" should be fixed
Line 424: :phenomenon" should be "phenomenal"
Excellent contribution.
Twice appears the word "belles," unless this is special use of the plural for bells, if should be fixed.
The sentence on lines 226-228 is awkward, the use of "motive."
The sentence on line 177, does the "may" fit with "will"?
Line 309: "these" should be "These
Line 356: needs end quote (")
377: "awaking" should be "awakening"
Line 412: "we" then "your" should be fixed
Line 424: :phenomenon" should be "phenomenal"
Author Response
Thank you for giving my paper a thorough review and for the insightful comments. I have made all the necessary changes to the manuscript based on your suggestions.
I further developed my engagements with Schaefer 1994 (lines 255-302), Gould et al. (lines 60-63, and 536-540) , and Faure 1991 (lines 809-811).
Reviewer 4 Report
This is an important article, which is well structured and eloquently written. It rightly argues for and indeed demonstrates the importance of complementing textual analysis with the study of the soundscape of Buddhist training centres. This is a significant step in developing a multifaceted phenomenology of embodied practice and sensual experience in the field of Buddhist Studies.
From the point of view of content, some of your comments on soundscape could benefit from incorporating reflection on Bernard Faure’s discussion of the ritualized nature of Zen meditation and on the importance of the Buddha (or patriarch as) icon. See Bernard Faure, The Rhetoric of Immediacy: A Cultural Critique of Chan/Zen Buddhism, Princeton: Princeton University Press, 1991, p.173, p.178, p.296, p.299, p.309.
ll.374–5: but also a means for spiritual transformation >> Might it perhaps be argued here that the landscape / soundscape of the Zen training temple is an extension of the master in his relationship with his disciples?
There are a few spelling errors and grammatical mistakes, which are noted below. As some of these recur many times, please make sure to check the article thoroughly for recurring typos, etc.
Also, there seems to be a problem with the numbering of the notes, as the number of endnotes does not correspond to those found in the article itself.
l.79: the traditional Buddhist sense. Because they >> the traditional Buddhist sense. This is because they
l.113: thirteen-century >> thirteenth-century
l.122: such as Eiheiji and Sōjiji, each has dozens >> change comma to semicolon
l.138: belles >> bells (check throughout)
l.144: and, as such, considered the locus classicus >> and, as such, is considered the locus classicus
ll.146–7: originally did not require elaborate >> originally it did not require elaborate
l.157: multiple percussion >> multiple percussions
l.168: he clearly views playing as a way >> playing instruments
l.169: as it’s approached >> as it is approached
l.177: [May] their hearing will perfectly purify them, >> either ‘may’ or ‘will’ need to be removed.
l.179: Boddhisatva >> bodhisattva (no need to capitalize when used as a generic term; note the correct spelling of the Skt. term.)
l.191: countless Buddhas >> do you need capitalization here?
l.208: belles >> bells
l.238: sutra >> sūtra (check throughout)
l.250: belles >> bells
l.291: sutra >> sūtra
ll.328–9: Fish-shaped Wooden drum >> please check your capitalization conventions; also where this term occurs above.
ll.342–6: quotation marks within a citation >> single quotation marks
l.356: from afar? >> from afar?” >> closing quotation marks
l.412: why do we put on your monk’s robe >> we or you?
l.424: phenomenon world >> phenomenal world
l.466: in religious and Buddhist >> delete ‘religious and’, as it’s unnecessary
l.507: Sweet anticipation: Music and the psychology of expectation >> words in English titles should be capitalized
Author Response
Thank you for giving my paper a thorough review and for the insightful comments. I have made all the necessary changes to the manuscript based on your suggestions. Please see the attached.

Round 2
Reviewer 1 Report
Revisions are now complete and publication of this article is recommended